# Occurrence, Concentration and Toxicity of 54 Polycyclic Aromatic Hydrocarbons in Butter during Storage

**DOI:** 10.3390/foods12244393

**Published:** 2023-12-06

**Authors:** Jianqiang Lan, Shimin Wu

**Affiliations:** Department of Food Science and Technology, School of Agriculture and Biology, Shanghai Jiao Tong University, 800 Dongchuan Road, Shanghai 200240, China; lanjianqiang@sjtu.edu.cn

**Keywords:** PAHs, butter, storage, oxidation, TEQs, POV

## Abstract

Polycyclic aromatic hydrocarbons (PAHs) are a class of highly carcinogenic compounds with a lipophilic nature. This study investigated the characterization of PAH24 contamination in twenty-one types of butter and five types of margarines using the QuEChERS pretreatment coupled with GC-QqQ-MS. Additionally, low-temperature storage experiments were conducted to explore the variations in oxidation index as well as the PAH levels. The results revealed that PAH24 concentrations in butter and margarine were 50.75–310.64 μg/kg and 47.66–118.62 μg/kg, respectively. The PAH4 level in one type of butter reached 11.24 μg/kg beyond the EU standards. Over 160 days of storage at 4 °C, acid value (AV), peroxide value (POV), and acidity significantly increased, while malondialdehyde (MDA) content and carbonyl value (CGV) fluctuated. Concentrations of PAH24 and oxidized PAHs (OPAHs) experienced a notable reduction of 29.09% and 63.85%, respectively. The slow reduction in naphthalene (NaP) indicated the dynamic nature of PAHs during storage. However, the toxic equivalency quotients (TEQs) decreased slightly from a range of 0.65–1.90 to 0.39–1.77, with no significant difference. This study contributes to the understanding of variations in PAHs during storage, which is of great significance for food safety.

## 1. Introduction

Polycyclic aromatic hydrocarbons (PAHs) constitute a distinctive category of organic compounds composed exclusively of carbon and hydrogen atoms, characterized by the presence of two or more benzene rings [1]. These compounds typically result from the processes of pyrolysis and the incomplete combustion of organic matter, which occur in both natural phenomena and human-induced activities, including the combustion of timber, petroleum, and gasoline [2]. The well-established ubiquity of PAHs in the environment and organisms has garnered significant scholarly attention based on their persistent nature, potential for bioaccumulation, and carcinogenic properties [3,4]. Furthermore, the subclassifications of oxidized PAHs (OPAHs) and halogenated PAHs (XPAHs) manifest augmented toxicological implications when compared to their parent PAHs (PPAHs) [5,6]. The substituent groups can increase carcinogenic, teratogenic, and mutagenic risks, while also enhancing mobility and bioavailability [7].

Given their lipophilic nature, PAHs possess a propensity to accumulate in foods characterized by elevated lipid content, and even within the food chain [8,9]. Butter provides noteworthy macro- and micronutrient contents and occupies an increasingly vital role in the human diet [10]. In moderation, butter offers nutritional benefits, providing healthy fats, fat-soluble vitamins, and minerals. With potential advantages for brain health and flavor enhancement, it can be a satisfying addition to a balanced diet. However, it is pertinent to acknowledge that butter, as a high-fat food source (fat content > 80%), exhibits a propensity for the accumulation of PAHs [11]. PAHs in raw milk can originate from contaminated feed, water, and grass consumed by cows [12,13]. Meanwhile, during the processing of raw milk into butter, pasteurization, homogenization, concentration, and other high-temperature procedures may contribute to PAH accumulation [14]. Notably, one study documented BaP concentrations as high as 3.29 µg/kg in butter, exceeding the established EU limit of 2 μg/kg in fats and oils [15]. Therefore, it is necessary to underscore the potential threat to human health resulting from its consumption.

Butter is frequently stored for extended durations [10]. However, studies on changes in PAHs in foods during storage are scarce and inconsistent. One study reported that both crude and refined vegetable oils could accumulate PAHs and OPAHs after 270 days of storage at 4 °C and 25 °C [16]. This phenomenon was attributed to the generation of cyclopentadienyl radicals, interactions among secondary oxidative products of fatty acids, and the auto-oxidation of unsaturated fatty acids. On the other hand, PAH levels and toxic equivalency quotients (TEQs) in yogurt continuously decreased during storage due to physical binding on the cell wall of fermented bacteria and molecular biological reactions [17]. No discernible variations in PAH levels were observed in fresh-cut potatoes, whether fried initially or after 2, 4, and 8 days of storage [18]. The presence of PAH contamination in butter during storage periods should not be overlooked.

There are limited data concerning PAH levels in butter in comparison to other dairy products such as milk [19], cheese [20], and yogurt [17], especially regarding changes during storage and contamination by PAH derivatives. To the best of our knowledge, this study stands as the first endeavor to monitor variations in PAH levels in commercial butter for the duration of its storage. The objectives are as follows: (i) to investigate the presence of PAH24 contamination in commercially available butter, (ii) to analyze changes in PAH54 concentrations during storage while concurrently evaluating relevant physicochemical attributes, and (iii) to assess potential exposure risks using the TEQs.

## 2. Materials and Methods

### 2.1. Chemicals

Fifty-four target PAH standards, which include 24 PPAHs, 18 XPAHs, and 12 OPAHs (>95%), were supplied by ANPLE Laboratory Technologies (Shanghai, China), Chiron Co., Ltd. (Trondheim, Norway), Dr. Ehrenstorfer (Augsburg, Germany), Cambridge Isotope Laboratories (Cambridge, UK), and Toronto Research Chemicals (Toronto, ON, Canada). The QuEChERS extraction salt packages, EMR-Lipid tubes, and polish tubes were obtained from Agilent Technologies (Santa Clara, CA, USA). Acetonitrile, acetone, and dichloromethane were of HPLC grades and sourced from Sinopharm Chemical Reagent Group Co., Ltd. (Shanghai, China). Ultrapure water was provided by a Milli-Q system (Millipore Co., Milford, CT, USA).

The standard solutions were dissolved in dichloromethane as a stock solution at a concentration of 1 mg/L in a mixed form and were stored at −20 °C, protected from light, in preparation for subsequent analysis.

### 2.2. Sample Preparation

Twenty-six commercial samples of different products in the first batch, including twenty-one types of butter (B) and five types of margarines (M), were collected from online shopping. These samples, from Belgium, China, France, Ireland, Netherlands, and New Zealand, were used to investigate the profiles of PAH24 contamination. Nutrition facts for them are shown in Appendix A.

After learning of PAH24 contamination in butter, another batch of fresh commercial butter, based on nine types with higher PAH concentrations, was purchased again to conduct further storage experiments and explore the changes in the levels of PAH54. These samples were numbered as AN*, AY, BN, BY, CN*, DN, DY, EN*, and EY. ABCDE is the brand, N stands for unsalted, Y stands for salted, and * indicates that the ingredient contains fermentation bacteria. The samples were stored at 4 °C for 160 days in the refrigerator to simulate natural storage conditions according to the manufacturer’s instructions. To prevent experimental interference, each butter was enveloped with plastic wrap and aluminum foil in addition to its original packaging.

### 2.3. PAHs Analysis

The determination of PAHs in butter was conducted by utilizing a QuEChERS pretreatment coupled with GC-QqQ-MS method previously used in our laboratory [6]. The butter specimen was melted at 60 °C in a water bath environment and then mixed thoroughly. A 2.0 g liquid butter sample was extracted by ultrasonication with 7.0 mL of ultrapure water and 10.0 mL of acetonitrile–acetone mixture (3:2, *v*/*v*) for 0.5 h. Next, the mixture was vigorously shaken with a package of QuEChERS extraction salt and centrifuged at 8000 rpm for 10 min. The resulting supernatant, 5.0 mL in volume, was transferred into an EMR-Lipid tube activated with 3.0 mL of ultrapure water in advance. After centrifuging at 10,000 rpm for 15 min, the supernatant was transferred into a polish tube. Following a subsequent centrifugation at 10,000 rpm for 20 min, a volume of 3.0 mL of the supernatant was concentrated to a mass of 20–40 mg using a mild stream of nitrogen and reconstituted to 250 μL with dichloromethane. Finally, the solution was subjected to filtration through a 0.22 μm membrane and preserved at −20 °C in the dark prior to GC-QqQ-MS analysis.

The DB-EUPAH column (20 m × 0.18 mm i.d. × 0.14 μm) was utilized for GC-QqQ-MS analysis (Agilent 7890B GC equipped with 7000D MSD; Agilent Technology, Santa Clara, CA, USA). The injection procedure employed a splitless mode, with an injection volume of 1.0 μL. The carrier gas was helium with a purity exceeding 99.999%, maintained at a constant flow rate of 1 mL/min. The temperatures of the injection port, transfer line, ion source and quadrupole were set at 280 °C, 300 °C, 230 °C and 150 °C, respectively. The temperature program was as follows: initially maintained at 80 °C for 1 min, raised to 180 °C at a rate of 20 °C/min, to 220 °C at a rate of 5 °C/min, to 280 °C at a rate of 3 °C/min, then further increased to 310 °C at a rate of 5 °C/min, and finally held for 10 min. The ion source was electron ionization (EI). Multiple reaction monitoring (MRM) mode was used for identification and quantification. Nitrogen, serving as the collision gas, was employed with a flow rate of 1.5 mL/min. The experimental conditions of MRM modes of GC-QqQ-MS for PAHs are shown in Appendix A. Each sample was analyzed at least twice in separate experiments. Quantifying the PAH concentrations in the samples involved the utilization of the Agilent QQQ Quantification Software B.09.00 (Santa Clara, CA, USA), relying on standard curves that were established externally. 

### 2.4. Oxidation Indicators

The acid value (AV), peroxide value (POV), and acidity were measured by the titration method in accordance with China National Standards GB 5009.229-2016 [21], GB 5009.227-2016 [22], and GB 5009.239-2016 [23], respectively. The malondialdehyde (MDA) content, also called the TBA value, and carbonyl value (CGV), were determined using the spectrophotometric method according to China National Standards GB 5009.181-2016 [24] and GB 5009.230-2016 [25]. It is worth noting that the butter should be melted in a 40 °C water bath condition in advance [26].

The oxidation stability index (OSI) was determined with a Metrohm Rancimat Model 892 (Herisau, Switzerland). In each test, 3.0 g of butter was used, the temperature was set at a constant 120 °C, and the flow rate of dry air was 20 L/h. OSI represents the time point on the water conductivity curve when the oxidation rate peaks [27,28].

### 2.5. Toxic Equivalency Factor (TEF) Method

TEQs can be calculated using various toxicity assessments, with the most commonly used approach being the TEF method. In this method [29], BaP is assigned a TEF of 1, and the toxicity of other PAHs is compared with that of BaP to determine their corresponding TEFs. The concentration of each PAH is then multiplied by its TEF to calculate its toxic equivalent. The sum of the toxic equivalents for all PAHs provides the total TEQs. These factors are determined through experiments and research.

### 2.6. Statistical Analysis

A statistical analysis of significant differences was performed using SPSS 27.0 software (Chicago, IL, USA). If the data followed a normal distribution, an independent *t*-test was conducted to compare the differences. If not, comparisons were made using the Mann–Whitney U test. Results were considered significant with a *p*-value < 0.5.

## 3. Results and Discussion

### 3.1. Concentrations of PAHs in Butter and Margarine

The analyses of 26 samples were performed in triplicate and the results are presented as the mean ± SD (Appendix A). The detected concentrations of PAH24 in 21 butter and 5 margarines ranged from 50.75 to 310.64 (mean value at 141.25) μg/kg and 45.74 to 118.62 (mean value at 72.01) μg/kg, respectively. Seventeen of the twenty-four PAHs were detected, with detection rates of NaP, Ap, Ac, F, Phe, Flu, Pyr, BcF, BaA, BbF, and BjF reaching up to 100.0%, and detections rates of BaP reaching 92.3%. Derived from the outcomes of quantification, the information for each PAH underwent normalization and was showcased through the depiction of a heatmap. As shown in the heatmap (Figure 1A), samples B11, B13, and B16 exhibited higher PAH concentrations, and DahA was only detected in sample B2. The high levels may be related to the high-temperature processing, the close proximity of the livestock to industrial settings, the environmental contamination of water, air, and the polluted feeds provided to the cows [30]. In this study, the levels of EPA 16 PAHs in butter were much higher than those in previous reports, with a mean value at 139.61 μg/kg, compared to 6.80 μg/kg [30], 19.13 μg/kg [31] and 72.80 μg/kg [15]. The lower levels in the first two reports may be due to the small number of brands and samples. The results of our study indicated that the PAH levels in butter may have been underestimated in previous reports and highlight the need for further extensive investigation. Regarding the compositions (Figure 1B), each set of 2-, 3-, and 4-ring PAHs in butter accounted for approximately one-third of the total PAHs, while in margarines, 2-ring PAHs contributed the most, followed by 4-ring and 3-ring PAHs. Nap, Phe, and Pyr exhibited relatively high levels among PAH24 in both butter and margarines. PAHs with three rings of samples B12 and B15 accounted for a substantial portion due to Phe.

As shown in Figure 2A, the concentrations of BaP and PAH4 (the sum of BaP, Chr, BaA, and BbF) in butter were 1.1 ± 0.4 μg/kg and 4.2 ± 2.6 μg/kg, respectively. In the case of margarines, the values were 0.8 ± 0.3 μg/kg and 3.0 ± 0.4 μg/kg. All samples met the EU standards (2 μg/kg and 10 μg/kg), although the PAH4 level of B11 was 11.24 μg/kg. The TEQs of butter and margarines were 0.42–2.29 (mean value at 1.54) and 0.85–1.59 (mean value at 1.23), respectively. Butter exhibited higher levels of PAH contamination and TEQs than margarines, which could be attributed to differences in raw materials and the preparation process. The concentrations of BaP exhibited a strong correlation with TEQs, as indicated by the Pearson correlation coefficient of 0.94 (Figure 2B). BaP remains the most suitable indicator of PAH-related health risks in butter. Moreover, the TEQ is computed with the BaP level, which serves as the reference point. This implies that butter’s TEQs level should be kept under 2. Utilizing this standard as the benchmark, four types of butter samples surpassed the prescribed limit. Considering the results presented above, there are valid concerns regarding PAH contamination in butter. In the future, it is imperative to enhance regulations concerning PAHs to encompass a broader range of food items. This includes consideration of other contaminants found in dairy products, such as melamine.

### 3.2. Changes of Oxidation Indices during Storage

The predominant issue associated with butter during storage pertains to rancidification, induced by lipolysis and oxidation of the fatty acids [10,32]. AV, also called the free fatty acid value (FFV), and POV reflect the degree of lipolysis and lipid oxidation, respectively. Oxidative reactions are characterized by chemical processes featuring a low activation energy, rendering them impervious to suppression through reductions in storage temperature. AV increased significantly after 160 days of storage, despite the volatility of the early period (Figure 3A). Except for sample DN, the average increase (0.19 mg/g) was close to that reported previously [10]. The changes in AV of butter during storage were mainly attributed to short-chain acids such as acetic acid, butanoic acid, and hexanoic acid [33]. Butter underwent relatively slight lipolysis due to the almost complete inactivation of milk lipoprotein lipase after pasteurization. The OSI in butter was 1.7 ± 0.3 h, and the DN sample showed the lowest OSI 1.01 h (Appendix A). In the later stages of storage, the AV of the DN sample rose rapidly, along with the off-flavors, and its surface turned dark, which led to failure to continue these experiments. All of these manifestations indicated that the butter was contaminated with microorganisms. This phenomenon was also true of another storage experiment we conducted at 25 °C. This may be relevant to the low OSI and ingredients without salt. Interestingly, microbial contamination did not significantly affect POV because the growth of the POV of DN was not as dramatic (Figure 3B). POV, which indicates primary oxidation, significantly increased from being undetected to 0.16–1.92 mmol/kg after 160 days. Notably, the POV of salted butter increased significantly faster than that of unsalted butter during storage, as determined by the Mann–Whitney U test. Due to the presence of metal impurities that reduced the activation energy, the addition of sodium chloride enhanced fat oxidation reactions, aligning with the reported results [10,34]. In comparison to unsalted butter, salted butter appeared to exhibit a higher level of safety risk and a decline in quality based on the changes in POV.

To further analyze the changes in the quality and secondary oxidation of butter during storage, the acidity, CGV, and MDA content were determined. The national standard of China limits the maximum acidity (GB 19646-2010) to 20 °T in butter (not applicable to products based on fermented cream) [35]. The acidity of all samples significantly increased after 160 days, especially for DN (Figure 4A). Both DN and DY exceeded the legislatively prescribed value. The quality of butter was greatly challenged after prolonged storage. However, CGV and MDA contents in butter fluctuated during storage (Figure 4B, C). This fluctuation could be related to the instability of secondary oxidation products, which can further degrade or combine with other substances [10]. MDA levels can be an indicator of oxidative stress and the extent of lipid peroxidation, which can affect the quality and shelf life of the butter. Monitoring MDA levels is a common practice in food quality control, especially in assessing the oxidative stability of fats and oils. However, there are no limits for CGV and MDA levels in butter in China at present.

### 3.3. Variations in PAH Concentrations in Butter during Storage

To conduct a more comprehensive examination of the alterations in PAHs throughout the storage period, PAH54 were detected in this study, including 24 PPAHs, 12 OPAHs, and 18 XPAHs. A total ion chromatogram of GC-QqQ-MS for the determination of PAH54 in sample AN* before and after storage is shown in Figure 5A. None of the XPAHs in butter were detected, possibly because of the low salt content [6]. The generation of PAHs has been extensively documented in high-temperature procedures like frying, baking, and grilling. Nevertheless, limited investigations have concentrated on the development of PAHs in foods at room temperature or in colder conditions. Most PAHs significantly decreased after 160 consecutive days of storage at 4 °C (Table 1). The *t*-test results illustrated the statistical significance of the reductions in most PAH levels observed both before and after storage. Most of the light PAHs were observed to decrease significantly after storage. The mean concentrations of OPAHs and PAH54 significantly decreased from 6.80 and 86.01 μg/kg to 2.46 and 58.63 μg/kg, respectively. In contrast, there were no significant changes in BaP, PAH4, PAH8, and EU PAH15+1 before and after storage. Both BaP and PAH4 were maintained at safe levels. There was no clear relationship between OPAHs and PAHs, and no clear evidence that POV and OPAHs were associated due to their opposite trends. 

This study revealed that a lower storage temperature contributes to a decrease in the risk of PAH contamination. Similar cases can be found in other foods. The levels of BaP, PAH8, and TEQ in yogurt decreased continuously, with maximum decrease rates of 56.44%, 68.32%, and 71.84% during 21 days of storage [17]. The concentration of EPA PAH16 in sheep casing and cellulose casing of smoked beef sausage reduced almost 100% from 2691 mg/kg and 599 mg/kg to 0.0026 mg/kg after 90 days of storage at 4 °C [36]. In contrast, Zhang et al. [16] reported a significant increase in soybean and rapeseed oils after 270 days of storage at 4 and 25 °C. The total increase was mainly attributed to NaP due to polymerization, oxidation, and radicals. Furthermore, Wang et al. [37] summarized that the different impact of storage on PAH levels in smoked foods and oils may result from the variant time sites of PAH formation, depending on whether heat is required.

Variations in PAH compositions in butter during storage are shown in Figure 5B. PAHs with three and four rings declined more than other PAHs, changing from 21.75 and 24.90 µg/kg to 10.08 and 16.13 µg/kg, respectively. The level of Phe in three-ring PAHs illustrated the largest decline, while NaP was still predominant among total PAHs. These results confirm that PAH levels in butter changed dynamically during storage, but demonstrated an overall decrease due to faster degradation than production. The presence of both together resulted in a much smaller decline in NaP. In addition, TEQs in butter decreased slightly from 0.65–1.90 to 0.39–1.77 after storage, although this difference was not significant (Figure 5C). This is mainly attributed to the chemically stable nature of highly toxic PAHs like BaP, which is the most appropriate indicator of PAH health risk in butter. The percentage of BaP in TEQs increased after storage due to the loss of total PAHs. In conclusion, the health risks associated with PAHs need to be taken seriously, even though PAH levels decreased during storage.

The decrease in PAH content in butter in this study may be due to natural losses, migration toward packaging, and utilization by lactic acid bacteria (LAB). Light PAHs are highly volatile compounds and are less stable and toxic compared to heavy PAHs [17]. As the oxidation levels rose during the later stages of storage, the heightened polarity of the butter intensified the repulsive interaction with nonpolar PAHs, thereby amplifying the volatility of PAHs. Another reason could be the migration of PAHs to packaging materials. As for the packaging material’s the removal effect on PAHs, Ciemniak and Kuźmicz [38] observed the following order: LDPE > HDPE > PET > glass, and absorption mainly took place during the initial 24 h of liquid media storage. Moreover, Ge and Wu [17] attributed the decline in PAH levels in yogurt to physical binding on the cell wall of LAB and molecular biological reactions depending on the pH, binding time, and temperature. The reasons underlying the decrease in PAHs in butter during storage are complex and not fully understood, requiring further extensive investigation.

## 4. Conclusions

This study revealed the profiles of PAH24 contamination in commercial butter and the changes in PAH54 levels during storage using the GC-QqQ-MS method combined with the QuEChERS pretreatment. The concentrations of BaP and PAH4 in butter were lower than the EU limit, except for one sample. Butter demonstrated higher PAH contamination and TEQ levels compared to margarine. AV, POV, and acidity in butter increased significantly during storage. Moreover, salted butter had significantly higher levels of POV than unsalted butter. PAH and OPAH levels significantly decreased during storage at 4 °C, suggesting that storage at a lower temperature is an effective means of reducing PAH contamination in butter. Notably, the reduction in the concentration of NaP was markedly slower than that observed for PAHs with three rings and four rings. This phenomenon underscores the dynamic and intricate nature of PAH levels over the storage period. However, the total TEQs did not change significantly. Early consumption of butter is recommended based on the changes in TEQs and quality during storage.

## Figures and Tables

**Figure 1 foods-12-04393-f001:**
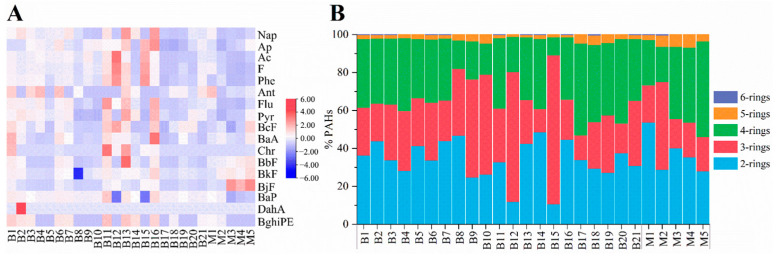
Heatmap (**A**) and ring distribution characteristics (**B**) of PAHs in 21 butter and 5 margarines.

**Figure 2 foods-12-04393-f002:**
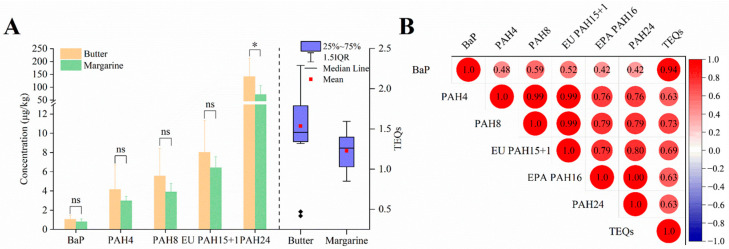
Concentrations (**A**) and Pearson correlation analysis (**B**) of PAH indicators and TEQs in butter and margarines. ns, no significance; *, *p* < 0.05; ◆, outliers.

**Figure 3 foods-12-04393-f003:**
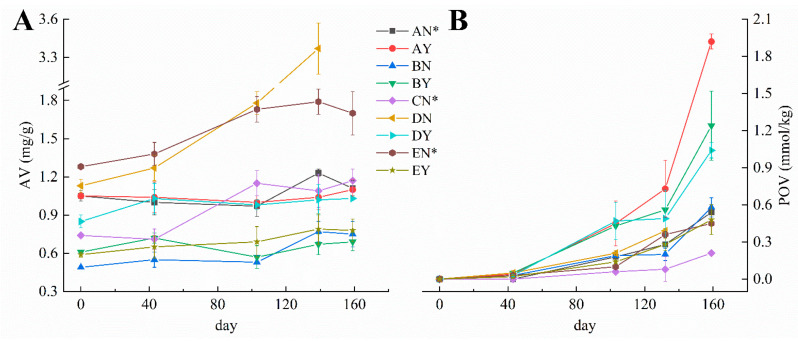
Changes in AV (**A**) and POV (**B**) in nine types of butter during storage.

**Figure 4 foods-12-04393-f004:**
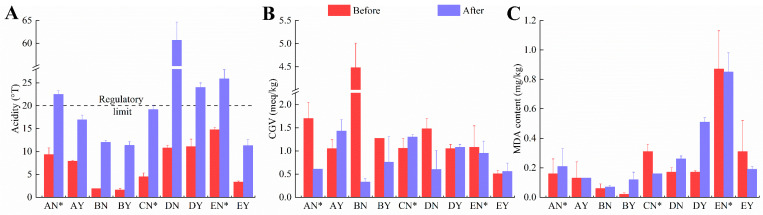
Changes in acidity (**A**), CGV (**B**), and MDA content (**C**) in nine types of butter before and after storage.

**Figure 5 foods-12-04393-f005:**
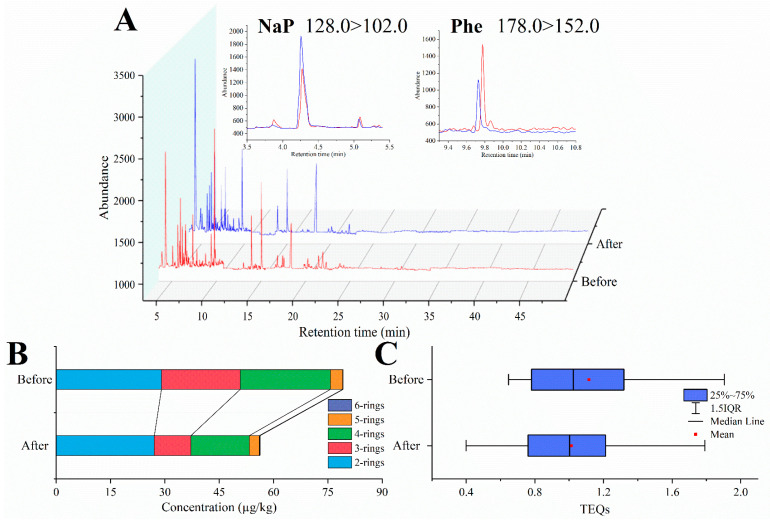
Total ion chromatogram for the determination of PAH54 in sample AN* (**A**); and changes in PAH compositions (**B**) and TEQs (**C**) in butter during storage.

**Table 1 foods-12-04393-t001:** PAH54 concentrations in butter before and after 160 days of storage at 4 °C.

Compounds	Before Storage (µg/kg)				After Storage (µg/kg)				
	Mean	Median	Min	Max	Mean	Median	Min	Max	
Naphthalene (Nap)	29.06	27.08	5.40	48.84	27.10	25.72	20.25	48.78	n.s.
Acenaphthylene (Ap)	0.63	0.50	0.33	1.77	0.32	0.30	0.25	0.44	*
Acenaphthene (Ac)	1.55	1.43	0.74	3.66	0.55	0.52	0.45	0.76	**
Fluorene (F)	2.71	2.44	0.99	5.61	1.21	1.19	0.79	2.06	**
Phenanthrene (Phe)	14.00	11.07	4.99	31.45	6.70	6.32	4.76	11.47	*
Anthracene (Ant)	2.85	2.70	0.82	4.83	1.30	1.23	0.81	2.04	**
Fluoranthene (Flu)	5.96	4.45	2.41	11.28	3.25	3.11	2.56	4.00	*
Pyrene (Pyr)	9.88	7.63	3.24	21.36	6.40	5.90	5.61	9.91	n.s.
Benzo[c]fluorene (BcF)	5.49	4.84	1.04	10.89	3.89	3.99	2.55	5.47	n.s.
Benz[a]anthracene (BaA)	1.53	1.48	0.67	2.84	0.95	0.95	0.55	1.51	*
Cyclopenta[c,d]pyrene (CP)	1.44	1.05	0.48	3.69	1.58	1.54	0.71	2.32	n.s.
Chrysene (Chr)	2.05	1.97	0.70	4.00	1.63	1.56	0.85	2.58	n.s.
Benzo[b]fluoranthene (BbF)	0.73	0.64	0.39	1.12	0.38	0.38	0.20	0.55	**
Benzo[k]fluoranthene (BkF)	0.34	0.30	0.06	0.78	0.14	0.13	0.05	0.24	*
Benzo[j]fluoranthene (BjF)	0.34	0.29	0.12	0.76	0.10	0.09	0.05	0.15	**
Benzo[a]pyrene (BaP)	0.41	0.38	0.17	0.77	0.54	0.43	0.08	1.00	n.s.
Dibenz[a,h]anthracene (DahA)	0.10	0.09	n.q.	0.28	0.04	n.q.	n.q.	0.15	*
Benzo[g,h,i]perylene (BghiPE)	0.13	0.12	0.02	0.36	0.08	0.07	0.01	0.19	n.s.
Benzanthrone (BZA)	0.17	0.16	0.06	0.38	0.06	0.04	0.01	0.12	**
1,4-Naphthoquinone (1,4-NQ)	0.02	n.q.	n.q.	0.11	0.06	0.02	n.q.	0.32	n.s.
1-Acenaphthenone(1-ANO)	0.14	0.13	0.05	0.35	0.07	0.07	n.q.	0.13	*
9-Fluorenone (9-FLO)	2.14	1.92	0.91	4.19	1.33	1.27	0.86	2.16	*
9,10-Anthraquinone (9,10-ATQ)	1.70	1.67	0.67	2.61	0.95	0.89	0.16	1.74	**
1,2-Acenaphthenequinone (1,2-ACQ)	1.52	1.50	0.62	2.38	n.q.	n.q.	n.q.	n.q.	***
2-Methyl-9,10-anthracenequinone (2M-ATQ)	1.11	1.09	0.60	1.98	n.q.	n.q.	n.q.	n.q.	***
PAH4	4.72	4.54	1.93	8.71	3.50	3.47	1.68	5.64	n.s.
PAH8	5.16	4.85	2.40	9.22	3.68	3.56	1.82	6.04	n.s.
EU PAH15+1	12.44	13.56	4.07	24.57	9.25	9.56	5.47	13.97	n.s.
EPA PAH16	71.80	63.18	25.96	105.38	50.52	44.86	40.49	71.25	*
OPAHs	6.80	6.41	2.96	11.89	2.46	2.45	1.34	4.39	**
PAH54	86.01	76.98	30.61	125.24	58.63	54.19	48.20	76.49	*

n.q., not quantified; n.s., no significance; *, *p* < 0.05; **, *p* < 0.01; ***, *p* < 0.001.

## Data Availability

Data is contained within the article or Appendix A.

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
