# Peer review of "Occurrence, Concentration and Toxicity of 54 Polycyclic Aromatic Hydrocarbons in Butter during Storage"

_foods, 2023, doi:10.3390/foods12244393_

Round 1

Reviewer 1 Report

Comments and Suggestions for Authors

The Authors investigate the content of PAHs compounds in dairy products including butter and margarine. The importance of monitoring such compounds in dairy products is duo to the fact that PAHs compounds in raw milk can be derived from contaminated feed, water, and grass consumed by cows. In addition, some manufacturing processes, especially at high temperature, such as pasteurization, homogenization, and concentration can contribute to the accumulation of PAHs in final products. Consequently, their monitoring is of fundamental importance to ensure the health of the consumers. However, this study is also focused on the exploration of variations of the PAHs content during the storage. In opinion of this Reviewer, such an approach increases the validity of the proposed manuscript giving valid and specific indications on the importance of storing food in appropriate conditions. However, it is not clear in the manuscript how the level of PAHs can variate during the storage period. Consequently, this Reviewer suggests to added in the introduction section a brief description regarding the reaction process that involve the variation of PAH contents in final products. In summary, in opinion of this Reviewer herein proposed manuscript, a minor revision is required.

 Some comments are listed below:

Abstract Section: “...using the QuEChERS-based GC-QqQ-MS method”. This Reviewer suggests modifying the sentence splitting the extraction protocol and gas chromatography-mass spectrometry method. Two procedures of extraction and analysis were carried in off-line, so there is not a direct connection.

Introduction Section pag 2 line 52: Please, specify the acronym “TEQs” in the text.

Materials and Methods Section pag 2 line 85: How was the PAH content established in nine different fresh commercial butters? Did the authors perform the analysis?

Materials and Methods Section pag 3 line 94: please, see same consideration described for the abstract section.

Materials and Methods Section pag 3 line 107-108: Please, specify the brand and model of the gas chromatograph system coupled to triple quadrupole mass spectrometer.

Materials and Methods Section pag 4 line 153-154: Please, report the type of collision gas used for the MRM experiments. In addition, how was the optimization of the MRM transitions done?

Results and Discussion Section pag 4 line 144: Please, change the sentence as follows: "the analyses of 26 samples was performed in triplicate...". The detection refers to analytes and no to samples.

Results and Discussion Section pag 4 line 148: PLease, specify the acronyms for all the compounds.

Results and Discussion Section pag 7 Figure 5: There is an inconsistency in Figure 5. The acquisition of the analytes was performed in MRM mode instead of scan mode as shown in Figure 5. Please, provide to modify the Figure 5 according to the MRM optimized transitions.

Results and Discussion Section pag 7 Table 1: This Reviewer suggests reporting the common name of compounds instead of name abbreviation. This could facilitate the lecture of the table.

Finally, this Reviewer suggests to report in the text an additional Table with optimized MRM transitions and relative gas collision energy. This could increse the visibility of the proposed manuscript.

Author Response

Journal: Foods

Manuscript ID: foods-2749563

Title: Occurrence, concentration and toxicity of 54 polycyclic aromatic hydrocarbons in butter during storage

Author(s): Jianqiang Lan; Shimin Wu.

Dear Editor

Many thanks for the constructive comments. The manuscript (ID: foods-2749563) has been amended according to the reviewer’s comments. Please see below for specific responses to each reviewer’s comments. We look forward to hearing from you. Thank you for considering our work for publication.

Moreover, we have added to the manuscript content and make sure the words up to 4000. The corresponding additions are as follows: “In moderation, butter offers nutritional benefits, providing healthy fats, fat-soluble vitamins, and minerals. With potential advantages for brain health and flavor enhancement, it can be a satisfying addition to a balanced diet” (Line 40-42); “Quantifying the PAH concentrations in the samples involved the utilization of the Agilent QQQ Quantification Software (Santa Clara, USA), relying on standard curves established externally” (Line 127-129); “Derived from the outcomes of quantification, the information for each PAH underwent normalization and was showcased through the depiction of a heatmap” (Line 162-164); “high-temperature processing, the close proximity of the livestock to industrial settings, environmental contamination of water, air, and the polluted feeds provided to the cows” (Line 166-168); “The results of our study indicated that the PAH levels in butter may have been under-estimated in previous reports and highlighted the need for further extensive investigation” (Line 171-173); “Moreover, the TEQ is computed with the BaP level serving as the reference point. This implies that butter's TEQs level should be kept under 2. Utilizing this standard as the benchmark, four types of butter samples surpassed the prescribed limit. Considering the results presented above, there are valid concerns regarding PAH contamination in butter. In the future, it is imperative to enhance regulations concerning PAHs to encompass a broader range of food items. This includes consideration for other contaminant found in dairy products, such as melamine” (Line 191-197); “Oxidative reactions are characterized by chemical processes featuring low activation energy, rendering them impervious to suppression through reduction of storage temperature” (Line 205-207); “as determined by the Mann–Whitney U test. Owing to the presence of metal impurities that reduced the activation energy” (Line 222-223); “In comparison to unsalted butter, salted butter appeared to exhibit a higher level of safety risk and a decline in quality based on the changes in POV” (Line 224-226); “MDA levels can be an indicator of oxidative stress and the extent of lipid peroxidation, which can affect the quality and shelf life of the butter. Monitoring MDA levels is a common practice in food quality control, especially in assessing the oxidative stability of fats and oils. However, there are no limits for CGV and MDA levels in butter in China at present” (Line 237-241); “The generation of PAHs has been extensively documented in high-temperature procedures like frying, baking, and grilling. Nevertheless, limited investigations have concentrated on the development of PAHs in foods under room temperature or colder conditions. The T-test results illustrated the statistical significance of the reductions in most PAH levels observed both before and after storage” (Line 250-256); “As the oxidation levels rose during the later stages of storage, the heightened polarity of the butter intensified the repulsive interaction with nonpolar PAHs, thereby amplifying the volatility of PAHs” (Line 293-296).

Reviewer 1’s comments:

The Authors investigate the content of PAHs compounds in dairy products including butter and margarine. The importance of monitoring such compounds in dairy products is duo to the fact that PAHs compounds in raw milk can be derived from contaminated feed, water, and grass consumed by cows. In addition, some manufacturing processes, especially at high temperature, such as pasteurization, homogenization, and concentration can contribute to the accumulation of PAHs in final products. Consequently, their monitoring is of fundamental importance to ensure the health of the consumers. However, this study is also focused on the exploration of variations of the PAHs content during the storage. In opinion of this Reviewer, such an approach increases the validity of the proposed manuscript giving valid and specific indications on the importance of storing food in appropriate conditions. However, it is not clear in the manuscript how the level of PAHs can variate during the storage period. Consequently, this Reviewer suggests to added in the introduction section a brief description regarding the reaction process that involve the variation of PAH contents in final products. In summary, in opinion of this Reviewer herein proposed manuscript, a minor revision is required.

 Response: Thank you for your comment. We have added a brief description regarding the reaction process that involve the variation of PAH contents in final products in the introduction section. “This phenomenon was attributed to the generation of cyclopentadienyl radicals, interactions among secondary oxidative products of fatty acids, and the auto-oxidation of unsaturated fatty acids” (Line 54-56) and “due to physical binding on the cell wall of fermented bacteria and molecular biological reactions” (Line 57-58).

Abstract Section: “...using the QuEChERS-based GC-QqQ-MS method”. This Reviewer suggests modifying the sentence splitting the extraction protocol and gas chromatography-mass spectrometry method. Two procedures of extraction and analysis were carried in off-line, so there is not a direct connection.

Response: Thank you for your comment. We have revised our manuscript to “using the QuEChERS pretreatment coupled with GC-QqQ-MS” in the Abstract section (Line 10-11), “utilizing a QuEChERS pretreatment coupled with GC-QqQ-MS method” (Line 100-101) and “using the GC-QqQ-MS method combined with the QuEChERS pretreatment” (Line 305-306).

Introduction Section pag 2 line 52: Please, specify the acronym “TEQs” in the text.

Response: Thank you for your comment. We have revised our manuscript to “and toxic equivalency quotients (TEQs) in yogurt” in the first mention of TEQs (Line 56-57), and “assess potential exposure risks using the TEQs” (Line 69).

Materials and Methods Section pag 2 line 85: How was the PAH content established in nine different fresh commercial butters? Did the authors perform the analysis?

Response: Thank you for your comment. We have added a back-and-forth order of relationships for ease of understanding. “After learning of PAH24 contamination in butter, another batch of fresh commercial butter based on nine types with higher PAH concentrations was purchased again…” (Line 90-91). Higher PAH concentrations were found in nine out of 21 types of butter. After that, we purchased these nine types of butter again. There were differences in before and after samples despite the same brands.

Materials and Methods Section pag 3 line 94: please, see same consideration described for the abstract section.

Response: Thank you for your comment. We have revised our manuscript to “using the QuEChERS pretreatment coupled with GC-QqQ-MS” in the Abstract section (Line 10-11), “utilizing a QuEChERS pretreatment coupled with GC-QqQ-MS method” (Line 100-101) and “using the GC-QqQ-MS method combined with the QuEChERS pretreatment” (Line 305-306).

Materials and Methods Section pag 3 line 107-108: Please, specify the brand and model of the gas chromatograph system coupled to triple quadrupole mass spectrometer.

Response: Thank you for your comment. We have specified the brand and model of the GC-QqQ-MS: “utilized for GC-QqQ-MS analysis (Agilent 7890B GC equipped with 7000D MSD; Agilent Technology, CA, USA)” (Line 115).

Materials and Methods Section pag 4 line 153-154: Please, report the type of collision gas used for the MRM experiments. In addition, how was the optimization of the MRM transitions done?

Response: Thank you for your comment. We have revised our manuscript to “The ion source was electron ionization (EI). Multiple reaction monitoring (MRM) mode was used for identification and quantification. Nitrogen, serving as the collision gas, was employed with a flow rate of 1.5 mL/min. Experimental conditions of MRM modes of GC-QqQ-MS for PAHs are shown in Table S2” (Line 122-126). Table S2 gives the optimization of the MRM transitions, such as RT (min), Quantitative ion (m/z) with Collision energy (eV), and Qualitative ion (m/z) with Collision energy (eV). The MRM transitions were optimized using PAH mixing standards, with various ionization energies selected to determine the optimal ionization energy for achieving the highest ion pair response for each PAH.

Results and Discussion Section pag 4 line 144: Please, change the sentence as follows: "the analyses of 26 samples was performed in triplicate...". The detection refers to analytes and no to samples.

Response: Thank you for your comment. We have revised our manuscript to “The analyses of 26 samples were performed in triplicate” (Line 157).

Results and Discussion Section pag 4 line 148: Please, specify the acronyms for all the compounds.

Response: Thank you for your comment. Considering that PAHs and their abbreviations were not mentioned in the first instance, we have added the corresponding full names in Table 1 for better readability. We checked the full text to ensure that the specific PAHs mentioned were abbreviated.

Results and Discussion Section pag 7 Figure 5: There is an inconsistency in Figure 5. The acquisition of the analytes was performed in MRM mode instead of scan mode as shown in Figure 5. Please, provide to modify the Figure 5 according to the MRM optimized transitions.

Response: Thank you for your comment. We have modified the Figure 5 according to the MRM optimized transitions. The acquisition of the analytes was performed in MRM mode.

Results and Discussion Section pag 7 Table 1: This Reviewer suggests reporting the common name of compounds instead of name abbreviation. This could facilitate the lecture of the table.

Response: Thank you for your comment. We have added the corresponding full names of compounds in Table 1 for better readability. We checked the full text to ensure that the specific PAHs mentioned were abbreviated.

Finally, this Reviewer suggests to report in the text an additional Table with optimized MRM transitions and relative gas collision energy. This could increase the visibility of the proposed manuscript.

Response: Thank you for your comment. We have added Table S2 that gives the optimization of the MRM transitions, such as RT (min), Quantitative ion (m/z) with Collision energy (eV), and Qualitative ion (m/z) with Collision energy (eV).

Now we have submitted the revised manuscript online. Thank you and the reviewers so much for the very helpful advices. We hope the revision meets the publishing criteria of the Foods and your approval.

Kind regards.

Yours sincerely,

Corresponding author: Shimin Wu

***********************************************

Prof. Wu, Shimin

Shanghai Jiao Tong University

Department of Food Science and Technology

School of Agriculture and Biology

P.O. Box 2-211, Agricultural Building

800 Dongchuan Road, Shanghai 200240

China

E-mail: wushimin@sjtu.edu.cn

Tel: +86-21-34205717

http://www.agri.sjtu.edu.cn/En/Data/View/2769

Reviewer 2 Report

Comments and Suggestions for Authors

The article entitled "Occurrence, concentration and toxicity of 54 polycyclic aromatic hydrocarbons in butter during storage " presents an interesting study on the evolution of HPA content during storage of butter and margarine. An adequate number of samples is studied by applying appropriate sample treatment and analytical methods.

However, some clarifications need to be made, which are presented below.

1.- Line 83: Indicate the countries of origin of the butter and margarine samples.

2.- Lines 85-86: How did the authors previously know that they had high PAH concentration?

3.- Table S2: Indicate correctly the significant figures. For example:

60.2 ± 0.1 instead of 60.19±0.10. 

6.8 ±0.3 instead of 6.76±0.26

This one is correct: 1.87±0.01

It is not correct to always put two decimal places. It depends on each case.

4.- Line 148: Somewhere in the article indicate the full names of the various PAHs. Not all readers need to know what each abbreviation means.

5.- Line 153: What do the authors want to indicate? An interval between 139 and 71 or is 139 the center point of an interval? It is not clear.

6.- Figures 1,2 y 4: The font size of the axes and legend is too small. They should be enlarged for better reading.

7.- Lines 165-166: Significant figures should be revised and modified throughout the text, not only in these two lines. When necessary only one decimal place. Not always two decimal places.

8.- Table 1, row 1, column 8: This minimum (20.25) is not in agreement with the associated mean and median. Why is this?

The article will be publishable after these minor revisions.

Author Response

Journal: Foods

Manuscript ID: foods-2749563

Title: Occurrence, concentration and toxicity of 54 polycyclic aromatic hydrocarbons in butter during storage

Author(s): Jianqiang Lan; Shimin Wu.

Dear Editor

Many thanks for the constructive comments. The manuscript (ID: foods-2749563) has been amended according to the reviewer’s comments. Please see below for specific responses to each reviewer’s comments. We look forward to hearing from you. Thank you for considering our work for publication.

Moreover, we have added to the manuscript content and make sure the words up to 4000. The corresponding additions are as follows: “In moderation, butter offers nutritional benefits, providing healthy fats, fat-soluble vitamins, and minerals. With potential advantages for brain health and flavor enhancement, it can be a satisfying addition to a balanced diet” (Line 40-42); “Quantifying the PAH concentrations in the samples involved the utilization of the Agilent QQQ Quantification Software (Santa Clara, USA), relying on standard curves established externally” (Line 127-129); “Derived from the outcomes of quantification, the information for each PAH underwent normalization and was showcased through the depiction of a heatmap” (Line 162-164); “high-temperature processing, the close proximity of the livestock to industrial settings, environmental contamination of water, air, and the polluted feeds provided to the cows” (Line 166-168); “The results of our study indicated that the PAH levels in butter may have been under-estimated in previous reports and highlighted the need for further extensive investigation” (Line 171-173); “Moreover, the TEQ is computed with the BaP level serving as the reference point. This implies that butter's TEQs level should be kept under 2. Utilizing this standard as the benchmark, four types of butter samples surpassed the prescribed limit. Considering the results presented above, there are valid concerns regarding PAH contamination in butter. In the future, it is imperative to enhance regulations concerning PAHs to encompass a broader range of food items. This includes consideration for other contaminant found in dairy products, such as melamine” (Line 191-197); “Oxidative reactions are characterized by chemical processes featuring low activation energy, rendering them impervious to suppression through reduction of storage temperature” (Line 205-207); “as determined by the Mann–Whitney U test. Owing to the presence of metal impurities that reduced the activation energy” (Line 222-223); “In comparison to unsalted butter, salted butter appeared to exhibit a higher level of safety risk and a decline in quality based on the changes in POV” (Line 224-226); “MDA levels can be an indicator of oxidative stress and the extent of lipid peroxidation, which can affect the quality and shelf life of the butter. Monitoring MDA levels is a common practice in food quality control, especially in assessing the oxidative stability of fats and oils. However, there are no limits for CGV and MDA levels in butter in China at present” (Line 237-241); “The generation of PAHs has been extensively documented in high-temperature procedures like frying, baking, and grilling. Nevertheless, limited investigations have concentrated on the development of PAHs in foods under room temperature or colder conditions. The T-test results illustrated the statistical significance of the reductions in most PAH levels observed both before and after storage” (Line 250-256); “As the oxidation levels rose during the later stages of storage, the heightened polarity of the butter intensified the repulsive interaction with nonpolar PAHs, thereby amplifying the volatility of PAHs” (Line 293-296).

Reviewer 2’s comments:

The article entitled "Occurrence, concentration and toxicity of 54 polycyclic aromatic hydrocarbons in butter during storage " presents an interesting study on the evolution of PAH content during storage of butter and margarine. An adequate number of samples is studied by applying appropriate sample treatment and analytical methods.

However, some clarifications need to be made, which are presented below.

1.- Line 83: Indicate the countries of origin of the butter and margarine samples.

Response: Thank you for your comment. We have added the countries of origin of the butter and margarine samples. “These samples, from Belgium, China, France, Ireland, Netherlands, and New Zealand” (Line 87-88).

 2.- Lines 85-86: How did the authors previously know that they had high PAH concentration?

Response: Thank you for your comment. We have added a back-and-forth order of relationships for ease of understanding. “After learning of PAH24 contamination in butter, another batch of fresh commercial butter based on nine types with higher PAH concentrations was purchased again…” (Line 90-91). Higher PAH concentrations were found in nine out of 21 types of butter. After that, we purchased these nine types of butter again. There were differences in the before and after samples despite the same brands.

3.- Table S2: Indicate correctly the significant figures. For example:

60.2 ± 0.1 instead of 60.19±0.10.

6.8 ±0.3 instead of 6.76±0.26

This one is correct: 1.87±0.01

It is not correct to always put two decimal places. It depends on each case.

Response: Thank you for your comment. We have revised our manuscript to correct the significant figures (Table S3).

4.- Line 148: Somewhere in the article indicate the full names of the various PAHs. Not all readers need to know what each abbreviation means.

Response: Thank you for your comment. Considering that PAHs and their abbreviations were not mentioned in the first instance, we have added the corresponding full names in Table 1 for better readability. We checked the full text to ensure that the specific PAHs mentioned were abbreviated.

5.- Line 153: What do the authors want to indicate? An interval between 139 and 71 or is 139 the center point of an interval? It is not clear.

Response: Thank you for your comment. Due to the large number brands of butter investigated in this study, PAH contents in butter were highly variable. We have revised our manuscript to “mean value at 139.61 μg/kg, compared to 6.80 μg/kg” (Line 169).

6.- Figures 1,2 y 4: The font size of the axes and legend is too small. They should be enlarged for better reading.

Response: Thank you for your comment. We have enlarged the font size of the axes and legend (Figures 1, 2, and 4).

7.- Lines 165-166: Significant figures should be revised and modified throughout the text, not only in these two lines. When necessarily only one decimal place. Not always two decimal places.

Response: Thank you for your comment. After checking the full text, we have revised our manuscript to “the concentrations of BaP and PAH4 (the sum of BaP, Chr, BaA, and BbF) in butter were 1.1 ± 0.4 μg/kg and 4.2 ± 2.6 μg/kg, respectively. In the case of margarines, the values were 0.8 ± 0.3 μg/kg and 3.0 ± 0.4 μg/kg” (Line 183-184) and “The OSI in butter was 1.7 ± 0.3 h” (Line 212).

8.- Table 1, row 1, column 8: This minimum (20.25) is not in agreement with the associated mean and median. Why is this?

Response: Thank you for your comment. Before storage, the levels of NaP in 9 types of butter were 27.08, 17.59, 5.40, 24.97, 31.41, 26.39, 31.13, 48.72, and 48.84 μg/kg, respectively. The minimum “5.40” shows much deviation from the whole and it could be an outlier in the experiment. After storage, they were 25.72, 25.89, 48.78, 22.00, 21.93, 30.67, 22.33, 20.25, and 26.34 μg/kg, respectively. The minimum (20.25) is in agreement with the associated mean (27.10) and median (25.72) (Table 1, row 1).

Now we have submitted the revised manuscript online. Thank you and the reviewers so much for the very helpful advices. We hope the revision meets the publishing criteria of the Foods and your approval.

Kind regards.

Yours sincerely,

Corresponding author: Shimin Wu

***********************************************

Prof. Wu, Shimin

Shanghai Jiao Tong University

Department of Food Science and Technology

School of Agriculture and Biology

P.O. Box 2-211, Agricultural Building

800 Dongchuan Road, Shanghai 200240

China

E-mail: wushimin@sjtu.edu.cn

Tel: +86-21-34205717

http://www.agri.sjtu.edu.cn/En/Data/View/2769